# Genome-Wide Association Analysis of Muscle pH in Texel Sheep × Altay Sheep F_2_ Resource Population

**DOI:** 10.3390/ani13132162

**Published:** 2023-06-30

**Authors:** Yilong Zhao, Sangang He, Jinfeng Huang, Mingjun Liu

**Affiliations:** 1College of Animal Science and Technology, Shihezi University, Shihezi 832000, China; ylzhao0823@sina.com; 2College of Animal Science and Technology, Xinjiang Agricultural Vocational and Technical College, Changji 831100, China; 3Biotechnology Institute, Xinjiang Academy of Animal Science, Urumqi 830013, China; 4Changji Huamu Animal Clinic, Changji 831100, China

**Keywords:** sheep, pH, SNP, 600 K BeadChip, whole genome resequencing, GWAS

## Abstract

**Simple Summary:**

pH was one of the main factors affecting meat storage/shelf life, and was also directly related to meat freshness, taste and overall quality. To extend storage/shelf life, pH-related candidate genes were mined to provide new opportunities for genetic improvement. We obtained candidate genes related to pH through a genome-wide association analysis to provide research directions for extending meat storage/shelf life.

**Abstract:**

pH was one of the important meat quality traits, which was an important factor affecting the storage/shelf life and quality of meat in meat production. In order to find a way to extend the storage/shelf life, the pH values (pH_45min_, pH_24h_, pH_48h_ and pH_72h_) of the longissimus dorsi muscles in F_2_ individuals of 462 Texel sheep × Altay sheep were determined, genotyping was performed using Illumina Ovine SNP 600 K BeadChip and whole genome resequencing technology, a genome-wide association analysis (GWAS) was used to screen the candidate genes and molecular markers for pH values related to the quality traits of mutton, and the effects of population stratification were detected by Q–Q plots. The results showed that the pH population stratification analysis did not find significant systemic bias, and there was no obvious population stratification effect. The results of the association analysis showed that 28 SNPs significantly associated with pH reached the level of genomic significance. The candidate gene associated with pH_45min_ was identified as the *CCDC92* gene by gene annotation and a search of the literature. Candidate genes related to pH_24h_ were *KDM4C*, *TGFB2* and *GOT2* genes. The candidate genes related to pH_48h_ were *MMP12* and *MMP13* genes. The candidate genes related to pH_72h_ were *HILPDA* and *FAT1* genes. Further bioinformatics analyses showed 24 gene ontology terms and five signaling pathways that were significantly enriched (*p* ≤ 0.05). Many terms and pathways were related to cellular components, processes of protein modification, the activity of protein dimerization and hydrolase activity. These identified SNPs and genes could provide useful information about meat and the storage/shelf life of meat, thereby extending the storage/shelf life and quality of meat.

## 1. Introduction

With the continuous improvement of people’s living standards, meat quality has attracted more and more attention from farmers and consumers. Farmers not only need to increase the yield of mutton, but also must constantly improve the quality of mutton. As one of the most important evaluation indexes of meat quality traits [1], pH is an important factor affecting the freshness and storage/shelf life of meat. Screening out the major genes affecting pH or molecular markers closely linked with them will provide theoretical support for extending the storage/shelf life of mutton. 

Since the application of the 600 K SNP commercial chip and whole genome resequencing in sheep, researchers in China and abroad have conducted genome-wide association analyses on various traits of sheep, such as the characteristics of the meat [2], reproductive traits [3,4], milk-producing traits [5,6], traits of wool quality [7,8], traits of growth and development traits [9,10,11,12,13] and traits of meat quality [14]. A large number of single-nucleotide polymorphism (SNP) markers associated with the target traits have been identified. So far, no GWAS analysis of the pH of mutton using a high-density SNP chip and whole-genome resequencing have been reported. Using a high-density SNP chip and whole gene resequencing for genotyping, combined with pH phenotype data for the GWAS analysis, more reliable results can be screened out.

Therefore, the objectives, in our study, are as follows: the pH values of the longissimus dorsi muscle at 45 min, 24 h, 48 h and 72 h after slaughter were measured. Next, the sheep 600 K chip technology and whole-genome resequencing technology were used to conduct a GWAS study in order to find the significant SNP sites that affected pH, and using the SNPs positions to identify candidate genes and pathways that may influence traits of pH. In addition, we will use the identified genes to conduct some bioinformatics analyses, such as gene ontology and the KEGG pathway for further analyses.

## 2. Materials and Methods

### 2.1. Animals 

The experimental animals were procured from the Sheep Breeding Experimental base of the Xinjiang Academy of Animal Science (Tianshan District, Urumqi City, China). In 2012, F_1_ individuals were produced using 3 pure-bred Texel sheep from New Zealand as the male parents and 220 Altay sheep as the female parents. Moreover, 4 F_1_ rams and 116 ewes were chosen to build the F_1_ population. From autumn 2014 to autumn 2019, 395 Crossbred F_2_ individuals and 67 backcrossed F_2_ individuals were produced by crossing (F_1_ rams × F_1_ ewes) and backcrossing (F_1_ raws × Altay ewes), using the simultaneous estrus and artificial insemination techniques with four F_1_ rams as the sires. After excluding individuals with incomplete information and abnormal development, 462 meat quality traits were scored and determined. SNP microarray genotyping was performed for 246 individuals, and whole-genome resequencing was performed for 216 individuals. Among them, 462 individuals of the F_2_ cross were developed as an F_2_ resource population managed by professional breeders. The conditions, forage materials and feeding methods used in each enclosure were the same for all six years. Samples from slaughtered animals were taken at the same time each year.

### 2.2. Sample Collection

All the experimental sheep were fed up to 8 months of age, and the live weight was measured before slaughter at the modern slaughterhouse of Hualing in Urumqi. Simultaneously, we sent the mutton to the dissection workshop for dissection of the carcass. Later, samples from the 12th to 13th ribs ofthe longissimus dorsi muscle on the left side of the carcass were collected to assess the traits of muscle pH. 

### 2.3. Phenotypic Determination and Correlation Analysis

Longissimus dorsi muscles from the 12th to 13th ribs of the carcass of Texel sheep × Altay sheep F_2_ individuals were collected. The pH was measured 45 min, 24 h, 48 h and 72 h after slaughter, and the samples were stored in a refrigerator at 4 °C between measurements. The pH was measured with a portable handheld pH meter purchased from TESTO, Germany. Each sample was measured 3 times, and the average was used as the final result [15]. In total, 462 individuals were tested for the phenotypic traits of pH_45min_, pH_24h_, pH_48h_ and pH_72h_. SPSS19.0 was used for the statistical analyses of the phenotypic data, and the correlation coefficients between the phenotypes were calculated.

### 2.4. Processing of the Genetic Data

Individuals of the F_2_ population of the Texel sheep × Altay sheep had ear tags applied within 24 h after birth. At the age of 2 months, tattoos were applied to the other ear with tattooing forceps, and the ear tags and tattoos were recorded. 

Meanwhile, ear tissue specimens were collected and placed in 2 mL preservation tubes (containing 75% ethanol), and then stored at −20 °C for future use. The genomic DNA of the ear tissue was extracted, and the concentration and purity of the samples were determined by a Nanodrop 2000 spectrophotometer. The qualified DNA samples were evenly diluted to 50 ng/μL. The DNA from the ear tissue of 462 qualified Texel sheep × Altay sheep F_2_ individuals were sent to Beijing Compson Biotechnology Co., Ltd. (Beijing, China), the DNA of 246 individuals was detected by Illumina Ovine SNP 600 K BeadChip and the DNA of 216 individuals was detected by whole genome resequencing (different sampling time was the main factor that caused different detection methods). Afterwards, the combination of chip data and resequencing data and the missing parts were filled with the genotype using beagle software. A total of 980,383 SNPs were obtained from 461 individuals. Subsequently, Plink1.9 [16] was used for quality control of the obtained genotypic data, and 613,178 SNPs were finally used for the analysis. The standards of quality control included the sample call rate, SNP call rate, minor allele Frequency (MAF) and other indicators, and the specific parameters are as follows: MIND < 0.05, GENO < 0.05, MAF > 0.05, HWE > 0.001. 

### 2.5. Phenotypic Variation Explained (PVE)

GCTA v1.93.2 software [17] was used to calculate the phenotypic variance of the significant SNPs’ interpretation screened by the genome-wide association analysis. The PVE was estimated by fitting a mixed linear model and using the maximum likelihood method. All the models were as follows:$$y=Wa+\sum_{i=1}^{r}g_{i}+\varepsilon$$
where *W* is the vector matrix of the fixed effect, α is the vector of the fixed effects, *g_i_* was the vector of the genetic effects, *g_i_* = $\sum_{i=1}^{m}z_{ij}u_{i}$, m is the significant number of SNPs, *u_i_* is the additive effect of the ith SNPs and *z*_ij_ is the correlation coefficient of the allele frequency. The estimation of PVE was equivalent to calculating the heritability of SNPs, namely $h_{snp}^{2}$ = $\sigma_{g}^{2}/\sigma_{p}^{2}$, where $\sigma_{g}^{2}$ was the total additive effect of variance and $\sigma_{p}^{2}$ was the phenotypic variance.

### 2.6. Population Structure Analysis 

In order to detect the population structure of the population samples, a principal component analysis was performed on the genotype data from 461 individuals. The GCTA v1.93.2 software [18,19] was used for the principal component analysis, and scatter plots of the first three principal components were drawn to show the distribution of the population structure. We observed the clustering of the samples, explored the influence of the genetic differences on the differences in the population and reduced the false positive caused by population stratification. The visualization of the results of PCA was performed using the R language.

### 2.7. Genome-Wide Association Analysis

GEMMA software was used for the GWAS analysis [20,21]. A mixed linear model (MLM) was used for the GWAS analysis of the phenotype of the pH values and genome-wide SNPs, and a Q–Q plot analysis was also performed.

The analytical model was:Y = Xα + Zβ + Wμ + e
where Y is the phenotypic trait, X is the indicator matrix of the fixed effects and α is the estimated parameter of the fixed effects. Z is the indicator matrix of the SNPs, β is the effect of the SNPs, W is the indicator matrix of the random effect, μ is the predicted random individual and e is the random residual, with e~ (0,δ e2). 

In the GWAS analysis, the effects of the population stratification effect can lead to false positive results [22]. It is necessary to determine whether a population is stratified by detecting the deviation between the distribution of the actual results of the experimental population and the distribution of the null hypothesis test. The results detected for the effects of population stratification are shown by Q–Q plots. In order to further test the likelihood of false positives in the association analysis, the genome inflation factor λ is calculated. When the expansion coefficient deviated from 1, this indicated that the population is more likely to be stratified [23], and the likelihood of false positives in the results is higher. 

### 2.8. Gene Annotation and Candidate Gene Data Mining for Candidate Genes

The sheep reference genome Ovis_aries_v3.1 was used to physically annotate the significant SNPs found in the GWAS and to identify candidate genes. 

The SNP databases of the UCSC and NCBI websites were used to determine the chromosomes and physical locations of the significantly-related SNP loci, and the ANNOVAR software [24,25,26] was used to annotate the genes within 2 Mb upstream and downstream from the significant SNP loci.

The standard databases were used for the Gene Ontology (GO) terms [27] and Kyoto Encyclopedia of Genes and Genomes (KEGG) pathway analysis [28], and the candidate genes related to the target traits were identified according to a study on the gene’s functions.

## 3. Results

### 3.1. Phenotypic Statistics Analysis

The pH values (pH_45min_, pH_24h_, pH_48h_ and pH_72h_) of the longissimus dorsi muscle samples of 462 F_2_ individuals collected from the experimental base from 2014 to 2019 were determined. The descriptive statistics included the mean, maximum, minimum and coefficient of variation. The phenotypic measurements of these traits were shown in Table 1. Among them, the mean of pH_45min_ was 5.98, that of pH_24h_ was 5.66, that of pH_48h_ was 5.63 and that of pH_72h_ was 5.47. Between 45 min and 72 h after slaughter, the pH value of the samples showed a decreasing trend at 4 °C; the pH decreasing trend was the same as that reported in the literature [29,30].

As can be seen from Table 2, a correlation analysis of the pH at various time periods after slaughter showed that there were no correlations among pH_45min_, pH_24h_, pH_48h_ and pH_72h_. There were significant positive correlations among pH_24h_, pH_48h_ and ph_72h_ (*p* < 0.01, r = 0.451 and *p* < 0.01, r = 0.458). There was a significant positive correlation between pH_48h_ and pH_72h_ (*p* < 0.01, r = 0.580).

### 3.2. Quality Control

After controlling the quality with Plink software, according to quality control standards, 461 individuals and 613,178 SNP loci were retained for the subsequent GWAS association analysis. See the Appendix A for the distribution and mean spacing of the SNP sites on each chromosome after quality control.

### 3.3. PVE

The PVE analyses of 28 sites significantly correlated with pH selected by GWAS were performed, and the results were shown in Table 3. The phenotypic contribution rate of five significant SNP sites for pH_45min_ was 0.3259. The phenotypic contribution rate of ten significant SNP sites for pH_24h_ was 0.2754. The phenotypic contribution rate of six significant SNP sites for pH_48h_ was 0.4452. The phenotypic contribution rate of seven significant SNP loci for pH_72h_ was 0.6129, and the phenotypic contribution rate of the significant loci in pH_72h_ was the highest among the four traits.

### 3.4. PCA

Based on the genomic data, PCA was performed on 613,178 SNP loci from all samples of the F_2_ population after quality control, and a cluster analysis was conducted using the first three principal components (PC1, PC2 and PC3). Each point in the PCA map represents a sample, and the aggregation of the grouped samples indicated that the differences among these samples were small, while a greater distance between groups indicated that the differences among these samples were large. Figure 1 shows the results. As can be seen from the figure, there were two subgroups in the studied population, which had obvious group stratification. To eliminate false positives due to population stratification, in this study, the first 10 principal components were extracted as covariables in the GWAS analysis model to reduce the false positives in the association analysis.

### 3.5. Analysis of Population Stratification

The results of detecting the effects of population stratification effect detection are shown in Figure 2. It can be seen that no clear overall systematic bias was found among the traits studied, and there was no obvious effect of population stratification, indicating that the results of this association analysis were highly reliable.

### 3.6. Genome-Wide Association Analysis

In the GWAS of pH, although no signal exceeded the Bonferroni significance threshold of GWAS (*p* < 8.15 × 10^−8^), a secondary GWAS signal was identified (*p* < 1 × 10^−5^) [31]. Thus, the determined significance threshold of the association was −log10 (*p*) >5. The GWAS results for pH are listed in Table 4 and Figure 3. The table gives the details of the significant SNP sites for pH in each time period, their positions on chromosomes and the *p*-values of the association. The QQMan software package of R was used to draw the Manhattan plots and visualize the results of the association analysis of pH in each time period [32].

In total, 28 SNPs significantly correlated with pH were detected by the genome-wide association analysis. Among these, five SNPs significantly correlated with pH_45min_ were located on chromosomes 1, 13, 17, 18 and 25; 10 SNPs significantly associated with pH_24h_ were located on chromosomes 2, 3, 8, 12 and 14; six SNPs significantly associated with pH_48h_ were located on chromosomes 3, 15, 20 and 23; and seven SNPs significantly associated with pH_72h_ were located on chromosomes 3, 4, 6, 16 and 26, respectively.

### 3.7. GO Function Analysis and KEGG Pathway Analysis

Using the UCSC Genome browser and NCBI database through the Asian server for the sheep reference genome Ovis_aries_v3.1, 36 genes were obtained within the region of 2 Mb up/downstream of the significant SNPs for pH traits.

Through the GO and KEGG enrichment analyses, we found 143 GO items for pH_45min_. There were 301 GO entries and 65 KEGG pathways for pH_24h_, among which 19 GO entries and 1 KEGG pathway were enriched significantly (*p* < 0.05). There were 244 GO terms and 17 KEGG pathways found for pH_48h_, among which 5 GO terms and 3 KEGG pathways were significantly enriched (*p* < 0.05). The results for pH_72h_ included 287 GO terms and 27 KEGG pathways, in which 1 KEGG pathway was significantly enriched (*p* < 0.05).

These GO terms and KEGG pathways were mostly associated with the regulation of the processes of protein modification, the activity of protein dimerization, hydrolase activity and the calcium signaling pathway; for details, see Figure 4, Figure 5, Figure 6 and Figure 7 and Table 5.

## 4. Discussion

### 4.1. pH Statisticals Results and Correlation Analysis

After slaughter, the blood circulation of sheep becomes stagnant, and the oxygen supply was insufficient. In this case, phosphocreatine phosphorylates ADP into ATP under the catalysis of phosphokinase to maintain the ATP level of the muscle cells [33]. Then, glycolysis provides energy for the muscle cells, and glycolysis produces pyruvate and ATP, which hydrolyzes and generates H^+^, and the H^+^ accumulates in the body, which reduces the pH [34,35]. Pyruvate generates lactic acid under the catalysis of lactate dehydrogenase, but the accumulation of lactic acid was not the main factor leading to muscle acidification. In vivo, lactic acid was produced at the same rate as H^+^ during glycolysis, so the content of lactic acid can reflect the muscle’s pH.

In this study, the mean value showed a downward trend from pH_45min_ to pH_72h,_ which was basically the same as the change in post-slaughter pH reported for other reported livestock and poultry [29,30,36]. Hamoen et al. [37] showed in their study that pH was a reliable indicator for predicting the postmortem quality of meat. The pH affects a variety of meat quality indexes by regulating protein activity. When the pH drops to the isoelectric point of muscle protein after slaughter, the protein denatures, which directly affects the muscle’s flesh color, tenderness, cooking loss, flavor, shelf life, preservation, etc. [38,39]. Postmortem muscle pH was not only affected by the muscle’s glycogen and lactic acid, but also regulated by the related metabolic enzymes such as AMp-activated protein kinase (AMPK) [40,41].

In this study, the pH at 45 min, 24 h, 48 h and 72 h after slaughter was determined. In the correlation analysis of pH at different time periods, there was no correlation of pH_45min_ with pH_24h_ to pH_72h_, which may be because glycolysis of the muscle had not started 45 min after slaughter, while pH_45min_ was related to the physiological status of the animals before slaughter [42].

There were significant positive correlations among pH_24h_, pH_48h_ and pH_72h_, which might be related to the activation of AMPK and glycolysis of the muscle after slaughter with the same postmortem time and the same storage environment [41,43,44].

### 4.2. Analysis of Significant Sites’ Rate and the Population Structure

The phenotype is the sum of the traits and characteristics exhibited by organisms under certain environmental conditions. Phenotypes are the result of a combination of the genotype and the environment. Genes and the environment are two main factors that affect biological traits. As internal and external factors, genes and the environment play their respective roles. Usually, biological traits are regulated by multiple genes, which are regulated by one or more pairs of genes in different formative periods. The genes interact with each other to affect the expression of traits.

In this study, based on the mixed linear model for single traits, GEMMA software was used to perform an association analysis for the pH traits. When the genomic significance criterion was −log10 (*p*) > 5, 28 significant loci were obtained, and the rate of phenotypic variation explained was 27.54–61.29%. In total, 36 genes were annotated from all loci. The rates of phenotypic variation explained at the significant sites were 32.59%, 27.54%, 44.52% and 61.29% for pH_45min_, pH_24h_, pH_48h_ and pH_72h_, respectively, indicating that more phenotypic variation was explained. These characteristics were regulated by genes and influenced by the environment.

In this study, the principal component analysis (PCA) was carried out on the two populations, and it was found that there was group stratification within the population, so the influence of group stratification should be considered in the analytical model. A mixed linear model of pH traits and phenotype values (MLM) was constructed, and a population stratification analysis was carried out using Q–Q plots. The results showed that the observed values and expected values were basically on a diagonal line, and the statistics of the observed values of significant SNP sites were above the expected values, indicating that the statistical distribution calculated by the SNP association analysis did not deviate from the hypothesis testing test. There was no group stratification in the population structure corrected by the kinship matrix, and the results of the mixed linear model of GEMMA were credible [20,21].

### 4.3. pH Function and Signal Pathway Analysis

According to the phenotypic traits forming four different pH periods, a GWAS analysis was conducted on the F_2_ population of Texel sheep × Altay sheep, and 28 SNP loci reaching the level of genomic significance were detected. Among the detected SNPs, 13 SNPs were located within the genes, 13 SNPs were located between the genes, 1 SNP was located in the exon and 1 SNP was located in the ncRNA_UTR5. The main reason for this phenomenon is that according to the whole genome reference sequence information of sheep (Ovis_aries_v3.1), the 600 K chip of sheep that was selected was mainly based on the SNP data of major sheep breeds around the world and lacked SNP data from Chinese sheep breeds crossed with foreign sheep breeds. There were many problems, such as the lack of homogeneity of the loci and the lack of representativeness of the functional loci and regions. Combined with the data from the whole-genome resequencing analysis, the accuracy of the results of this experiment were improved compared with using the chip data alone.

The gene ontology enrichment analysis and KEGG pathways, in our study, revealed a number of GO terms and KEGG pathways. These involved significant genes related to the traits under examination; for example, the regulation of the protein modification process was a term (GO:0031399) of pH_24h_ in the biological process category, which has four genes (Figure 5). Among them, *TGFB2* was considered to be the nearest gene to our significant SNPs; *TGFB2* and *GOT2* were enriched in multiple terms, including the establishment of localization (GO:0051234), the obsolete multi-organism process (GO:0051704) and the organonitrogen compound metabolic process (GO:1901564). The cellular component organization was a term (GO:0016043) of pH_48h_ in the biological process category, which has six genes (Figure 6). Among them, *MMP12* and *MMP13* were considered to be the nearest genes to our significant SNPs; *MMP12* and *MMP13* were enriched in multiple terms, including the obsolete multi-organism process (GO:0051704) and the cellular component organization or biogenesis (GO:0071840). The cell morphogenesis was a term (GO:0000902) of pH_72h_ in the biological process category, which has three genes (Figure 7). Among them, *FAT1* was considered to be the nearest gene to our significant SNPs; *FAT1* was enriched in terms of cellular component morphogenesis (GO:0032989) and *HILPDA* and *FAT1* were enriched in terms of cell-cell signaling (GO:0007267).

The protein dimerization activity was a term of pH_24h_ (GO:0046983) in the molecular function category, which has four genes (Figure 5). *TGFB2* and *GOT2* were considered to be the nearest genes to our significant SNPs; *GOT2* was enriched in terms of enzyme binding (GO:0019899) and *TGFB2* was enriched in terms of molecular function regulator activity (GO:0098772). The activity of hydrolase was a term of pH_48h_ (GO:0016787) in the molecular function category, which has three genes (Figure 6). *MMP13* and *MMP12* were considered to be the nearest genes to our significant SNPs; *MMP13* and *MMP12* were enriched in terms of enzyme binding (GO:0043169) and the metal ion binding (GO:0046872). The cation binding was a term of pH_72h_ (GO:0043169) in the molecular function category, which has one gene (Figure 7). *FAT1* was considered to be the nearest genes to our significant SNPs; *HILPDA* and *FAT1* were enriched in terms of binding (GO:0005488).

This above functional analysis suggested that some genes might be involved in the changes of muscle pH after slaughter through proteolytic degeneration [45,46].

The top three KEGG analysis pathways of pH_24h_ were map00220: arginine biosynthesis, map00250: alanine aspartate and glutamate metabolism, and map00270: cysteine and methionine metabolism, all of which contain a GOT2 gene. The top three KEGG analysis pathways of pH_48h_ were mapmap04657: IL-17 signaling pathway, map04926: relaxin signaling pathway, and map04928: parathyroid hormone synthesis secretion and action, all of which include MMP13 and MMP12 genes.

The functional analysis yielded many terms and pathways related to proteolytic activity, protein regulatory repair and the biological process. Therefore, it is reasonable to presume that all significant SNPs and candidate genes might be associated with pH.

### 4.4. Analysis of the Candidate Gene of pH_45min_

Through the genome-wide association analysis, the SNP loci significantly associated with pH_45min_, pH_24h_, pH_48h_ and pH_72h_ ofthe longissimus dorsi muscle were detected, which were located on chromosomes 1, 2, 3, 4, 6, 8, 12, 13, 14, 15, 16, 17, 18, 20, 23, 25 and 26. Three SNPs were significantly correlated with pH_24h_, pH_48h_ and pH_72h_, and all of them were located on chromosome 3.

According to the genomic location of SNP loci associated with a significant trait, the significant loci were annotated by ANNOVAR software. One SNP (OAR17_51353307) located on chromosome 17 among the SNPs significantly correlated with pH_45min_ was located in the 5 ‘UTR region of the non-coding RNA of *CCDC92*. *CCDC92* is a coiled-coil domain protein, and a member of the coiled coil domain protein family, which mainly includes structural proteins, regulatory proteins, enzymes, regulatory factors, etc. A common feature of CCDC proteins is that they contain a curly helix domain with a superhelix structure formed by two to five helices encircling each other. The curly helix domain of the CCDC protein was used for many purposes, which enables the protein to have various functions [47,48]. It has been reported that *CCDC92* was an unknown molecular substance that affects adipocyte differentiation [49]. Lotta et al. [50] showed that *CCDC92* affects the expression of fat genes, thus impairing adipogenesis, reducing peripheral fat storage and increasing the risk of heart disease. Klarin et al. [51] showed that *CCDC92* was significantly correlated with body fat percentage, and triglyceride and adiponectin levels, and a correlation was found with lipid metabolism disorders.

Based on the results of the above literature, *CCDC92* may regulate the process of fat metabolism in the muscle and affect the muscle’s pH_45min_. Therefore, it was speculated that *CCDC92* can be used as a candidate gene to affect pH_45min_.

### 4.5. Analysis of the Candidate Gene of pH_24h_

Two SNPs (OAR2_74191733 and OAR2_74212523), which were located on chromosome 2, were among the SNPs significantly correlated with pH_24h_ and were all located on the intron of *KDM4C*. *KDM4C* (lysine-specific demethylase 4C, *KDM4C*) was also known as *GASC1* (Gene amplified in squamous cell carcinoma 1) or *JMJD2C* (Jumonji domain containing 2C). *KDM4C* was a histone demethylase belonging to the histone lysine demethylase (*KDM*) family. Larson et al. [52] showed in their study that a high-fat (HF) diet in mice after birth had a greater effect on the obesity and the number of beige fat cells (BA) in the offspring than a high-fat (HF) diet during pregnancy. Maternal exercise induced *KDM4C* to counter-increase G9a (histone 3 lysine 9 dimethyltransferase) induced by the HF diet after birth, and the increased expression of the G9a protein reduced the number of beige fat cells (BA). In their study, Zhang et al. [53] used GEMMA and FarmCPU to conduct a GWAS on seven traits and identified 82 SNPs. Two QTLS associated with body type traits were detected on SSC8 and SSC17 by both methods. It was found that *TNFAIP3*, *KDM4C*, *HSPG2*, *BMP2*, *PLCB4* and *GRM5* were candidate genes related to the body weight and body shape of pigs.

Two SNPs (OAR12_20003596 and OAR12_20013208) located on chromosome 12 were all located on the intron of *TGFB2*. *TGFB2* was a subunit of transforming the growth factor-β (*TGF-β*), which has a certain influence on the growth and development of muscle. *TGF-β* was a superfamily of proteins that play an important role in the regulation of cell growth [54] and differentiation [55]. Li et al. [56] found that *TGF-β* has a widespread influence on regulating the growth, development and body composition of chickens. Ueda et al. [57] showed that *TGF-β* was a growth factor that can induce the secretion of *COL4A* in connective tissue. Compared with subcutaneous adipose tissue, the expression of *TGFB2* and *TGFB3* was higher in intramuscular adipose tissue, while the expression of *TGFB1* was lower. In addition, an affymetrix microarray was used to analyze the gene expression profiles of abdominal adipose tissue of diabetic (GK) and control (WKY) rats. An important gene that was highly expressed in GK rats was the transforming growth factor β2 (*TGFB2*) [58], and the increased expression of *TGFb2* inhibited lipogenesis [59].

A SNP (OAR14_25937544) located on chromosome 14 was close to the *GOT2* gene and was 46.92 kb away from the gene. *GOT2* (glutamic-oxalotransaminase 2, *GOT2*) was a type of glutamic-oxaloacetic transaminase 2 (*GOT*) in eukaryotes and exists in the cell’s mitochondria. Studies have shown that *GOT* was involved in amino acid metabolism and the tricarboxylic acid cycle (TCA), and it plays a role in cell proliferation. Decreased *GOT2* can inhibit cell proliferation and slow down the rate of glycolysis [60].

In conclusion, *KDM4C* and *TGFB2* regulate muscle pH by affecting adipocyte and lipogenesis. The *GOT2* gene affects muscle pH through its participation in muscle glycolysis, so the above three genes can be used as candidate genes to influence pH_24h_.

### 4.6. Analysis of the Candidate Gene of pH_48h_

One SNP (OAR15_5271378) on chromosome 15 among the SNPs significantly correlated with pH_48h_ was located between *MMP13* and *MMP12*. *MMP13* (matrix metalloproteinase 13, *MMP13*) was one of the important members of the matrix mettalloproteinases (*MMPs*) family, and its main function was to degrade the extracellular matrix. Smith et al. have shown that *MMP13* was essential for muscle growth and regeneration [61]. Lei et al. [62] found that matrix metalloproteinases (*MMPs*) played a key role in the remodeling of the extracellular matrix (ECM) during muscle regeneration. In the muscles of chronically injured muscular dystrophy mice (mdx mice), the expression of *MMP13* and its protein levels were elevated, and in differentiated mouse myoblast C2C12 cells, the expression of *MMP13* was most pronounced after fusion of the myoblasts and during the formation of the myotube, suggesting that *MMP13* plays a major role in the migration of myoblasts.

*MMP12* (matrix metalloproteinase 12, *MMP12*), also known as macrophage elastase, was secreted mainly by macrophages, monocytes and other inflammatory cells as a kind of endoprotease. *MMP12* can decompose components of the extracellular matrix and vascular components. Meanwhile, *MMP12* was closely related to adipose tissue. The literature has shown that *MMP12* can promote fibrosis in the process of repairing muscle injuries in rats [63,64]. Chung et al. found that UV irradiation could induce the expression of the *MMP12* gene and protein in human skin and fibroblasts [65]. Jing et al. found that *MMP12* plays a key role in controlling the weight of APCMin/+ mice [66]. Huang et al. [67] showed that in samples from head and neck cancer (HNC) patients, the expression of *DRP1* was positively correlated with the expression of *FOXM1* and *MMP12*. Deletion of *DRP1* affected aerobic glycolysis by down-regulating glycolysis genes, and the overexpression of *MMP12* in cells with a deletion of *DRP1* helped restore the consumption of glucose and the production of lactic acid.

In line with the results of the literature shown, it was found that *MMP12* may affect muscle pH through the process of repairing muscle injuries and the glycolysis pathway. *MMP13* may affect muscle pH through the processes of muscle growth and regeneration, so it was speculated that *MMP12* and *MMP13* may be candidate genes affecting pH_48h._

### 4.7. Analysis of the Candidate Gene of pH_72h_

Among the SNPs significantly correlated with pH_72h_, a SNP (OAR4_92689824) located on chromosome 4 was adjacent to the gene *HILPDA*, 17.22 kb away from this gene. *HILPDA* (hypoxia-inducible lipid droplet-associated, *HILPDA*) is also known as *C7orf68*, *HIG-2* and *HIG2*. Proliferator-activated receptors (*PPARs*) were the target of peroxisome proliferators, which participate in the secretion of triglycerides and regulate the metabolism of fat in the liver [68]. The literature has shown that *HILPDA* regulates the cellular metabolism of lipids [69,70,71], cellular responses to hypoxia [72] and the homeostasis of cellular lipid droplets [73].

A SNP (OAR26_15183899) located on chromosome 26 was located in the exon region of *FAT1*. *FAT1* (Fat atypical cadherin 1) is a member of the fat cadherin superfamily and is a transmembrane protein of about 500 kDa, which has the functions of regulating the dynamics of actin, intercellular adhesion, cell polarity, etc. *FAT1* is a group of transmembrane proteins commonly expressed in the epithelial tissues, with functions in adhesion molecules and/or signal transduction receptors [74,75]. The FAT1 protein was involved in the formation of cell–cell adhesion, and interference with the expression of *FAT1* can reduce the stability of cell connections and destroy the cells’ polarity [76]. Lai et al. showed that the content of omega-3 polyunsaturated fatty acids in transgenic Fat1 pigs was significantly higher than that in non-transgenic pigs [77]. Pan et al. [78] studied the effect of knocking out isocitrate dehydrogenase (*IDH2*) on the homeostasis of the energy of skeletal muscles and showed that the expression levels of adipogenic pathway genes (Pparg, Znf423 and Fat1) were downregulated in *IDH2* knockout (KO) mice. Meanwhile, mitochondrial NADP+-dependent isocitrate dehydrogenase (*IDH2*) catalyzes the oxidative decarboxylation of isocitrate to α-ketoglutaric acid and the reduction of NADP+ to NADPH. *IDH2* acts as a metabolic regulator in the tricarboxylic acid cycle (TCA) by catalyzing the conversion of alpha-ketoglutaric acid, and as a major REDOX regulator through the production of NADPH.

According to the results from the literature presented above, it was found that *HILPDA* may affect muscle pH through regulating the metabolism of fat. *FAT1* may affect muscle pH through the fatty acid metabolism pathway and through participating in the process of adipogenesis, etc. Therefore, *HILPDA* and *FAT1* are speculated to be candidate genes affecting pH_72h_.

In this study, GWAS was used to preliminarily screen candidate genes related to the pH of sheep meat. To determine the functions of potential candidate genes or loci, functional verification of the cells and populations should be further conducted by molecular biological methods, so as to determine the influence of the candidate genes or loci on the traits of meat quality, to better guide the improvement of local breeding resources.

## 5. Conclusions

According to four phenotypic traits of pH in the F_2_ population of Texel sheep × Altay sheep at different time periods, 28 SNPs were identified by GWAS. Finally, *CCDC92* was identified as the candidate gene for pH_45min_, *KDM4C*, *TGFB2* and *GOT2* as the candidate genes for pH_24h_, *MMP12* and *MMP13* as the candidate genes for pH_48h_, and *HILPDA* and *FAT1* as the candidate genes for pH_72h_. The main processes of the above genes were identified, namely protein modification and hydrolase activity. Our findings provide useful information for understanding the changes in the pH of mutton after slaughter and the storage/shelf life of meat, and they provide a new research direction for improving the storage time and meat quality of mutton.

## Figures and Tables

**Figure 1 animals-13-02162-f001:**
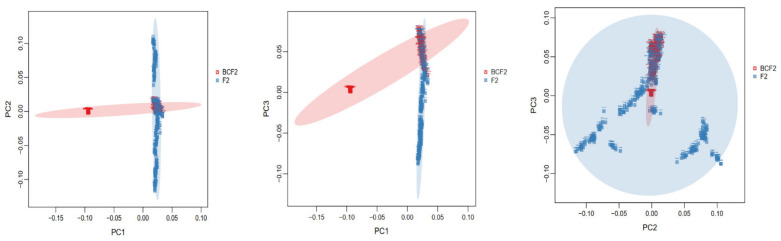
PCA plot of the principal component analysis. The population in this study was composed of BCF_2_ and F_2_, with obvious stratification. In order to eliminate false positives caused by population stratification, the top 10 principal components were extracted in this study as covariables in the GWAS analysis model to reduce false positives in the association analysis.

**Figure 2 animals-13-02162-f002:**
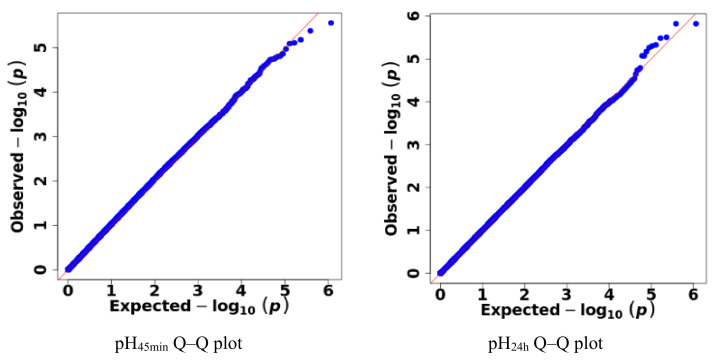
Quantile–quantile (Q–Q) diagram of the effects of population stratification. Q–Q plot showing the late separation between observed and expected values. The red lines indicate the null hypothesis of no true association. Deviation from the expected *p* value distribution is evident only in the tail area for different pH traits, indicating that population stratification was properly controlled.

**Figure 3 animals-13-02162-f003:**
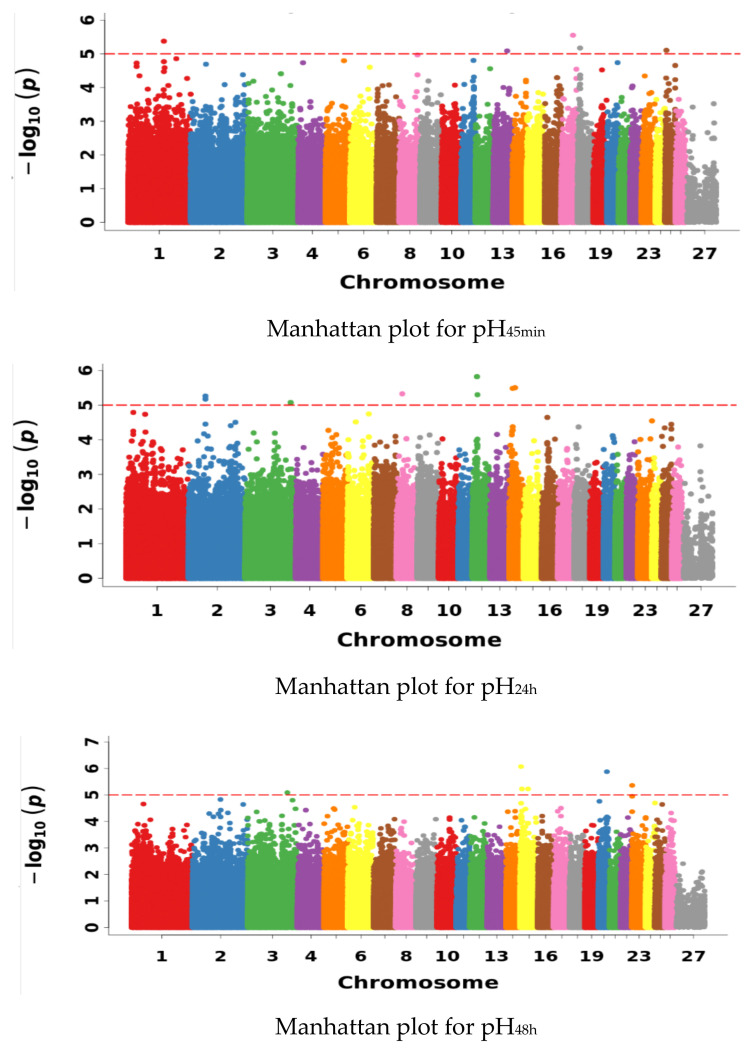
Manhattan Plots of pH in different time periods. Manhattan plot in which the genomic coordinates of SNPs are displayed along the horizontal axis, the negative logarithm of the association *p* value for each SNP is displayed on the vertical axis and the red dotted line indicates the significance threshold level after the secondary recognition signal (secondary signal recognition when no signal is recognized after the Bonferroni correction).

**Figure 4 animals-13-02162-f004:**
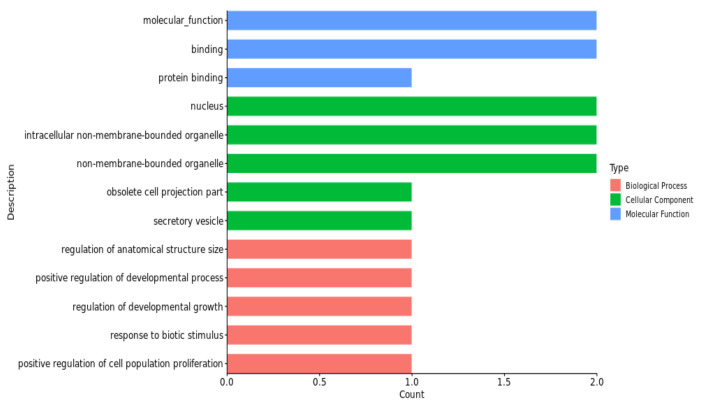
Significant GO terms (biological process, cellular component and molecular function) of candidate genes related to the pH_45min_ trait.

**Figure 5 animals-13-02162-f005:**
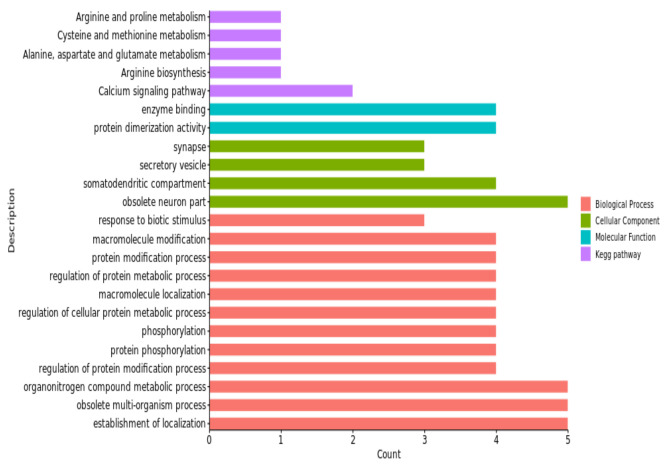
Significant GO terms (biological process, cellular component and molecular function) and KEGG pathways of candidate genes related to the pH_24h_ trait.

**Figure 6 animals-13-02162-f006:**
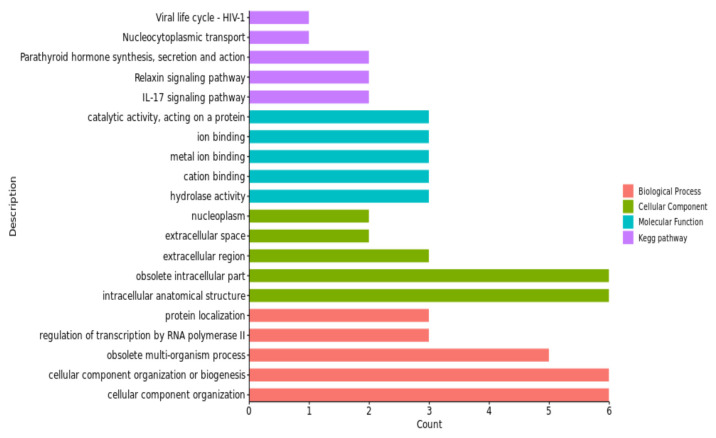
Significant GO terms (biological process, cellular component and molecular function) and KEGG pathways of candidate genes related to the pH_48h_ trait_._ In the pH_48h_ candidate gene enrichment analysis and KEGG pathway analysis, the top five enrichment items were selected for mapping.

**Figure 7 animals-13-02162-f007:**
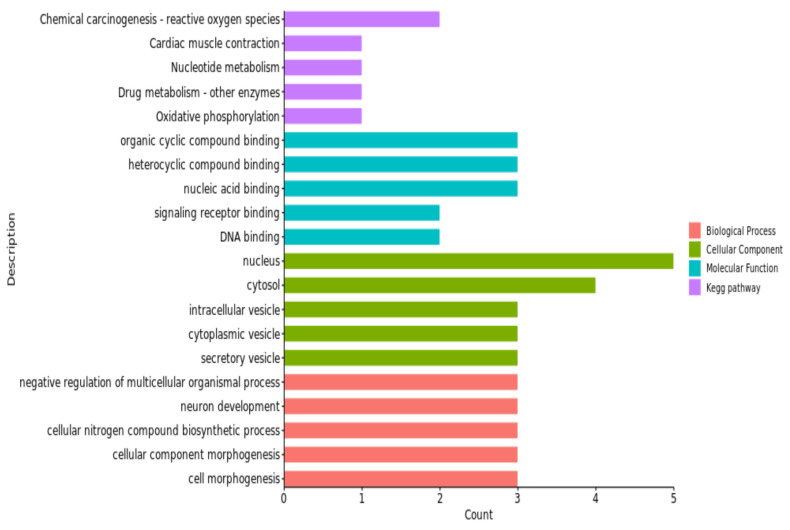
Significant GO terms (biological process, cellular component and molecular function) and KEGG pathways of candidate genes related to the pH_72h_ trait. In the pH_72h_ candidate gene enrichment analysis and KEGG pathway analysis, the top five enrichment items were selected for mapping.

**Table 1 animals-13-02162-t001:** pH ofthelongissimus dorsi muscle in the Texel sheep × Altay sheep F_2_ population.

Trait	Individuals (n)	Mean	Standard Deviation	Maximum	Minimum	Coefficient of Variable
pH_45min_	461	5.98	0.14	7.05	5.54	2.34
pH_24h_	462	5.66	0.17	6.66	4.61	3.00
pH_48h_	462	5.63	0.18	6.57	5.02	3.20
pH_72h_	461	5.47	0.17	6.65	4.54	3.11

pH_45min_, pH of sheep at 45 min after slaughter; pH_24h_, pH of sheep at 24 h after slaughter; pH_48h_, pH of sheep at 48 h after slaughter; pH_72h_, pH of sheep at 72 h after slaughter.

**Table 2 animals-13-02162-t002:** Correlations among the phenotypic traits of pH.

	pH_45min_	pH_24h_	pH_48h_	pH_72h_
pH_45min_	1			
pH_24h_	0.126 **	1		
pH_48h_	−0.073	0.451 **	1	
pH_72h_	0.001	0.458 **	0.580 **	1

pH_45min_, pH of sheep at 45min after slaughter; pH_24h_, pH of sheep at 24 h after slaughter; pH_48h_, pH of sheep at 48 h after slaughter; pH_72h_, pH of sheep at 72 h after slaughter. The superscript ** represents a significant correlation at 0.01.

**Table 3 animals-13-02162-t003:** Contribution rate of significant sites for pH.

Trait	SNP	Genetic Variance (V_G_)	Residual Variance (V_E_)	Phenotypic Variance (V_P_)	Phenotypic Variation Explained (V_G_/V_p_)
pH_45min_	5	0.0335	0.0693	0.1029	0.3259
pH_24h_	10	0.0374	0.0983	0.1358	0.2754
pH_48h_	6	0.0646	0.0805	0.1451	0.4452
pH_72h_	7	0.0714	0.0452	0.1165	0.6129

pH_45min_, pH of sheep at 45 min after slaughter; pH_24h_, pH of sheep at 24 h after slaughter; pH_48h_, pH of sheep at 48 h after slaughter; pH_72h_, pH of sheep at 72 h after slaughter. V_G_/V_p_ indicates the value of phenotypic variation explained (PVE).

**Table 4 animals-13-02162-t004:** SNP sites significantly correlated with pH.

Trait	Chromosome	Location ^a^	*p*-Value	Nearest Gene Name ^b^	Distance
pH_45min_	1	156943874	4.21 × 10^−6^	*NSUN3*	within
	13	60319420	8.17 × 10^−6^	*LOC101123139*, *LOC105611319*	50 kb
	17	51353307	2.81 × 10^−6^	*CCDC92*	within
	18	10181225	6.72 × 10^−6^	*LOC105603046*, *LOC105603047*	100 kb
	25	4816881	7.86 × 10^−6^	*DISC1*	within
pH_24h_	2	74191733	6.71 × 10^−6^	*KDM4C*	within
	2	74212523	5.48 × 10^−6^	*KDM4C*	within
	3	200982308	8.44 × 10^−6^	*GRIN2B*	within
	3	200982636	8.44 × 10^−6^	*GRIN2B*	within
	8	24764513	4.71 × 10^−6^	*LOC105615845*, *LOC105609596*	1 Mb
	12	20003596	1.51 × 10^−6^	*TGFB2*	within
	12	20013208	1.51 × 10^−6^	*TGFB2*	within
	12	22099281	5.02 × 10^−6^	*RAB3GAP2*, *MARK1*	50 kb
	14	15539367	3.29 × 10^−6^	*PHKB*	within
	14	25937544	3.15 × 10^−6^	*GOT2*, *LOC105611954*	50 kb
pH_48h_	3	178286132	8.17 × 10^−6^	*LOC105608627*, *TRNAR-CCU*	1 Mb
	15	5271378	8.53 × 10^−6^	*MMP13*, *MMP12*	100 kb
	15	9099535	5.94 × 10^−6^	*CNTN5*	within
	15	36890512	5.93 × 10^−6^	*LOC105602233*, *INSC*	100 kb
	20	38742466	1.33 × 10^−6^	*NUP153*	within
	23	680144	4.36 × 10^−6^	*LOC105604591*	within
pH_72h_	3	76238035	1.29 × 10^−6^	*LOC105613977*, *LOC101113750*	200 kb
	3	76238358	1.29 × 10^−6^	*LOC105613977*, *LOC101113750*	200 kb
	4	25686591	4.23 × 10^−6^	*LOC105609497*, *AHR*	100 kb
	4	92689824	5.14 × 10^−6^	*IMPDH1*, *HILPDA*	50 kb
	6	65142470	3.96 × 10^−6^	*GABRA2*, *LOC101121518*	500 kb
	16	39886318	4.49 × 10^−6^	*ADAMTS12*	within
	26	15183899	7.72 × 10^−6^	*FAT1*	within

^a^. Position of the SNPs on Ovis_aries_V3.1, ^b^. Annotated according to Ovis_aries_V3.1.

**Table 5 animals-13-02162-t005:** Partial results of the GO terms and KEGG pathway analysis.

Gene Name	Term	Database	ID	*p*-Value
*GRIN2B*, *TGFB2*, *RAB3GAP2*, *MARK1*	Regulation of the processes of protein modification.	Gene Ontology	GO:0031399	0.010966194
*GRIN2B*, *TGFB2*, *RAB3GAP2*, *GOT2*	Activity of protein dimerization.	Gene Ontology	GO:0046983	0.011199421
*GRIN2B*, *PHKB*	Calcium signaling pathway.	KEGG pathway	map04020	0.035660879
*MMP13*, *MMP12*, *INSC*, *NUP153*, *LOC105604591*	Obsolete multi-organism process.	Gene Ontology	GO:0051704	0.042182781
*MMP13*, *MMP12*, *CNTN5*, *INSC*, *NUP153*, *LOC105604591*	Cellular component organization.	Gene Ontology	GO:0016043	0.038462380
*MMP13*, *MMP12*, *LOC105604591*	Hydrolase activity.	Gene Ontology	GO:0016787	0.027560590
*MMP13*, *MMP12*	Parathyroid hormone synthesis, secretion and action.	KEGG pathway	map04928	0.035660879

The GO terms and KEGG analysis for the partial candidate genes of pH_24h_ and pH_48h_ with genome-wide significant association.

## Data Availability

The data presented in this study are available on request from the first author or the corresponding author.

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
