# Peer review of "Genome-Wide Association Analysis of Muscle pH in Texel Sheep × Altay Sheep F2 Resource Population"

_animals, 2023, doi:10.3390/ani13132162_

Round 1

Reviewer 1 Report (Previous Reviewer 2)

Thank you for addressing issues presented in the earlier version. Here are suggestions to improve the quality of this paper.

* Please explain the pathways and their significance in the discussion.

* Are those markers being in LD?

* Needs revision on sentence structure. Significant topic and presentation can be improved.

* What are the proportion of variance explained by those individual markers

* Please add more details in table and figure legend/description.

Author Response

Dear reviewer

Thanks for your valuable comments on the article. Due to my limited English level and scope of knowledge, there may be some problems in the process of revision. Please timely feedback to me, I will continue to revise according to your requirements, thanks again for your hard work! Comments on modification are attached. The revised manuscript has been sent to you.

Kind regards,

Yilong Zhao

Response to Reviewer 1 Comments

Point 1: Please explain the pathways and their significance in the discussion.

Response 1: According to the suggestions of experts, the approaches have been discussed in the article and the meaning has been clarified.

Point 2: Are those markers being in LD?

Response 2: Sites significantly correlated with pH contributed more to the phenotype and belonged to the primary locus. However, other microefficent sites did not meet the significance standard due to repeated detection and correction, so the significant sites found were not all in LD.

Point 3: Needs revision on sentence structure. Significant topic and presentation can be improved.

Response 3: According to the expert's advice, the sentence pattern has been checked and modified in the article, but due to the limited level, it may not meet the requirements, but I will try my best to meet the requirements.

Point 4: What are the proportion of variance explained by those individual markers

Response 4: The proportion of variance explained by these individual markers can be found at

Table 3(Contribution rate of significant site for pH). 

Point 5: Please add more details in table and figure legend/description.

Response 5: According to the experts' suggestions, I made detailed supplements to the diagrams and other instructions by referring to the articles published by Animals.

Reviewer 2 Report (New Reviewer)

Simple summary & abstract clearly present the aim of the study and practical value of it.

Key words list should be more specific.

line 41 citation is needed

lines 43-44 this part should be wilder described, with data representing mutton meat.

line 53 - it shouldn't be the main reason of conducting studies. Please reformulate the sentence.

Introduction part is quite superficial - reviewer advice re-writting it.

M&M part please avoid duplication of the information.

line 86 carcass separation

lines 87& 90 presenting different length of cut

What says the lwa regulation about storage of fresh meat in China? why Authors have chosen 4oC?

what were detailed conditions of storage? bacteria load? and the feeding is important information - influencing on meat quality; animal condition? BCS?

slaughter&carcass breakdown conditions?

any organoleptic analysis of the meat - in the reviewer's opinion it should be conducted

line 294 - not proved by the Authors

Figures legends are very brief, must be improved

lines 384-385 LD representing lean meat - and the Authors didn't provide  details of the diet

lines 402-403/418-419 not important for the research

Please avoid comparisons with cancer examples/tumors - not relevant for the analysis

Any confirmation of gene expression differences?

Its difficult to analyze Figs. 4 -7

Limitations of the study?

Ethical 

the overall quality of the language used in the manuscript is fine

Author Response

Dear reviewer

Thanks for your valuable comments on the article. Due to my limited English level and scope of knowledge, there may be some problems in the process of revision. Please timely feedback to me, I will continue to revise according to your requirements, thanks again for your hard work! Comments on modification are attached. The revised manuscript has been sent to you,Please see the attachment .

Kind regards,

Yilong Zhao

Response to Reviewer 2 Comments

Point 1: Simple summary & abstract clearly present the aim of the study and practical value of it.

Response 1: According to the opinions of experts, the Simple summary and abstract are supplemented to clarify the significance of this study.

 Point 2: Key words list should be more specific.

Response 2: According to the suggestions of experts, the keywords are adjusted and the keywords that can represent the full text are listed as keywords.

 Point 3: line 41 citation is needed

Response 3: Here you can refer to Li Fan's thesis (Fan Li. Spatio-temporal evolution of livestock and poultry meat production and its influencing factors in China, Lanzhou: Lanzhou University, 2022, 5. ). In China, there are many reports on the ranking of mutton production by provinces, so there are no annotated references.

 Point 4: lines 43-44 this part should be wilder described, with data representing mutton meat.

Response 4: This part has been deleted or rewritten in the article, I hope experts can understand

 Point 5: line 53 - it shouldn't be the main reason of conducting studies. Please reformulate the sentence.

Response 5: According to the expert's suggestion, the meaning of the sentence has been supplemented and perfected in the article.

 Point 6: Introduction part is quite superficial - reviewer advice re-writting it.

Response 6: According to the opinions of experts, the introduction part has been rewritten by referring to the previous writing methods of related articles published by Animals.

 Point 7: M&M part please avoid duplication of the information.

Response 7: Thanks to the expert's reminding, the repeated parts have been integrated, which makes the article smoother.

 Point 8: line 86 carcass separation

Response 8: When the carcass was divided, it was used for the determination of meat traits, while the longmuscle samples were used to determine the meat quality traits of the sheep, including muscle pH.

 Point 9: lines 87& 90 presenting different length of cut

Response 9: Thanks to the expert's discovery, when I started the determination of meat quality traits, I carried out a large number of testing items and a large number of samples. I did not pay attention to the consistency with the content of this paper when writing here, resulting in inconsistency.

 Point 10: What says the lwa regulation about storage of fresh meat in China? why Authors have chosen 4℃?

Response 10: Meat preservation See Synonyms at GB/T 40464-2021《Technical requirements for processing of chilled meat》. In this paper, the research was carried out according to the national standard of chilled meat, mainly to ensure the consistency of the storage environment of samples, so the 4℃ storage environment was selected in this study.

 Point 11: what were detailed conditions of storage? bacteria load? and the feeding is important information - influencing on meat quality; animal condition? BCS?

Response 11:

  1. The samples of longus dorsi muscle were stored in a glass dish and stored in a special 4℃ During the measurement, the glass dish for storing samples was placed on broken ice to minimize the number of opening and closing the refrigerator, so as to keep the refrigerator temperature constant and ensure the same environment during the measurement.
  2. The glass dishes containing meat samples have been sterilized, and the storage environment is the same. Under the condition of 4℃, bacteria will breed, but it will not affect the experimental results. I hope the experts can understand that we did not measure the content of bacteria on the surface of meat samples, but only ensure the same storage and testing conditions.
  3. Sheep feeding in accordance with Chinese Agricultural Industry Standards (NYT816-2021 Nutrient requirements of meat-type sheep and goat).

d.The experimental animal feed comes from forage farming farms around the experimental base, and the water quality is not polluted. Other added feeds ensure the same feeding population. Feeding animal products may not meet the BCS

Point 12: slaughter&carcass breakdown conditions?

Response 12: Slaughtering and carcass segmentation are mainly carried out to meet all the experiments of the project team. Meat properties are determined before carcass segmentation, tissue samples such as back longus muscle, half tendon muscle, half membrane muscle and back fat are collected after segmentation, and the remaining carcasses are sold after passing quarantine inspection. In this study, the longus dorsi muscle was used as the sample.

 Point 13: any organoleptic analysis of the meat - in the reviewer's opinion it should be conducted

Response 13: We conducted phenotypic determination and analysis of 11 meat traits in the research, but the whole-genome association analysis of muscle pH is the most innovative and significant, so we choose to submit the article of whole-genome association analysis of pH to MDPI "Animals".

Point 14: line 294 - not proved by the Authors

Response 14: I hope the experts can understand. Perhaps due to the limited amount of literature I searched, I did not find any reports of pH correlation. I only analyzed the possible causes of pH correlation

Point 15: Figures legends are very brief, must be improved

Response 15: According to the opinions of experts, the Figures legends part is supplemented and modified by referring to the related articles published by Animals in the past.

 Point 16: lines 384-385 LD representing lean meat - and the Authors didn't provide  details of the diet

Response 16: In this study, sheep were fed according to Chinese agricultural industry standards. Reference 54 was cited to prove that KDM4C can regulate adipocytes.

Point 17: lines 402-403/418-419 not important for the research

Response 17: Thanks to the expert's reminding, irrelevant content and references have been deleted from the article.

 Point 18: Please avoid comparisons with cancer examples/tumors - not relevant for the analysis

Response 18: Cancer-related content and references have been deleted, thanks to expert advice.

Point 19: Any confirmation of gene expression differences?

Response 19: In this study, no control group was set up in the design of the experiment, so no differentially expressed genes were found. However, transcriptome analysis of meat quality traits of different breeds of sheep will be conducted in the newly applied scientific research project, and the differentially expressed genes controlling this trait will be found, which I hope experts can understand.

 Point 20: Its difficult to analyze Figs. 4 -7

Response 20: In this paper, the obtained genes were analyzed by Go trem and kegg, and the four maps were combined into one map to make it more clear. The top 5 entries in the functional enrichment analysis and pathway analysis were selected into the diagram. For clarity, I will add an explanation of the chart to the article.

 Point 21: Limitations of the study?

Response 21: The phenotypic data used in this paper only included the data of 462 individuals. The more phenotypic data, the better the reliability of the correlation analysis results.

 Point 22: Ethical 

Response 22: The study protocols for rearing and slaughtering the animals were reviewed and approved by the guidelines (1 December 2014) of the Institute of Biotechnology, Xinjiang Academy of Animal Science, and Xinjiang Academy of Animal Science.

Reviewer 3 Report (Previous Reviewer 1)

This is an interesting manuscript aimed to identify genes associated with muscle pH in a Texel x Altay sheep population. However, there are still some suggestions that should be considered to improve the manuscript:

-       Line 25: Separate the sentences typing a space in between.

-       Lines 55-59: The paragraph should be single-spaced.

-       Line 75: Separate the sentences typing a space in between.

-       Line 89: I suggest replacing “statistical analysis” by “correlation analysis”.

-       Line 107: Replace the “comma” by a “period”.

-       Line 108: Replace “tissue DNA” by “DNA tissues”.

-       Line 109: Separate the sentences typing a space in between.

-       Lines 111, 112: Separate the round bracket from the text.

-       Lines 116-118: I suggest clarifying to which quality criterion each parameter corresponds.

-       Line 133: What is the reference of the software?

-       Line 156: Why the study did not include a test to adjust for multiple SNP comparisons such as Bonferroni adjustment test?

-       Line 162: What is the reference of the software?

-       Line 269: Separate the number from the text.

-       Line 272: Replace “is” by “was”.

-       Line 293: Replace “correlation” by “association”.

-       Line 326: Separate the square bracket from the text.

-       Line 346: Replace “the” by “The”.

-       Line 351: Separate the square bracket from the text.

-       Line 354: Correct the font size of the sentence “the longissimus dorsi muscle”.

-       Line 356: Remove the word “respectively”.

-       Lines 379, 381, 382, 394, 414, 430, 445, 476: Separate round brackets from the text.

-       Line 511: Replace “Conclusion” by “Conclusions”.

-       Lines 512-519: This paragraph seems more like a results summary than conclusions.

-       Line 531: In References section the volume number should be in italic style.

Author Response

Dear reviewer

Thanks for your valuable comments on the article. Due to my limited English level and scope of knowledge, there may be some problems in the process of revision. Please timely feedback to me, I will continue to revise according to your requirements, thanks again for your hard work! Comments on modification are attached. The revised manuscript has been sent to you,Please see the attachment .

Kind regards,

Yilong Zhao

Response to Reviewer 3 Comments

Point 1: The following grammar problems

-Line 25: Separate the sentences typing a space in between.

-Lines 55-59: The paragraph should be single-spaced.

-Line 75: Separate the sentences typing a space in between.

-Line 89: I suggest replacing “statistical analysis” by “correlation analysis”.

-Line 107: Replace the “comma” by a “period”.

- Line 108: Replace “tissue DNA” by “DNA tissues”.

-Line 109: Separate the sentences typing a space in between.

-Lines 111, 112: Separate the round bracket from the text.

-Lines 116-118: I suggest clarifying to which quality criterion each parameter corresponds.

-Line 269: Separate the number from the text.

-Line 272: Replace “is” by “was”.

-Line 293: Replace “correlation” by “association”.

-Line 326: Separate the square bracket from the text.

-Line 346: Replace “the” by “The”.

-Line 351: Separate the square bracket from the text.

-Line 354: Correct the font size of the sentence “the longissimus dorsi muscle”.

-Line 356: Remove the word “respectively”.

-Lines 379, 381, 382, 394, 414, 430, 445, 476: Separate round brackets from the text.

-Line 511: Replace “Conclusion” by “Conclusions”.

-Line 531: In References section the volume number should be in italic style.

Response 1: According to the opinions of experts, all the revisions have been made in the article

 Point 2: -Line 133: What is the reference of the software?

Response 2: Thanks to the expert's reminding, the reference for the application of this software has been marked in the article.

 Point 3: -Line 156: Why the study did not include a test to adjust for multiple SNP comparisons such as

Bonferroni adjustment test?

Response 3: If Bonferroni test is used, P=0.05/613178 = 8.15e-08 is the threshold value, and there is no significant site in the data in this study. So refer to GWAS related literature(https://doi.org/10.1186/s13059-019-1648-9)to reduce the threshold to 1e-5, that is, -log10 (P) >5.

Point 4: -Line 162: What is the reference of the software?

Response 4: Thanks to the expert's reminding, the reference for the application of this software has been marked in the article.

Point 5: Lines 512-519: This paragraph seems more like a results summary than conclusions.

Response 5: Thanks to the careful and meticulous experts, the conclusion has been adjusted.

Round 2

Reviewer 2 Report (New Reviewer)

Thank you for following my suggestions during article revision.

Please check language once more.

Please check language once more. Little mistyping errors still existing

This manuscript is a resubmission of an earlier submission. The following is a list of the peer review reports and author responses from that submission.

Round 1

Reviewer 1 Report

This is an interesting manuscript aimed to screen genomic candidate genes for pH values related to quality traits of mutton in Texel sheep x Altay sheep. I suggest to clarify some important issues related to GWAS analyses and consider several minor comments.

Abstract. According to “Instructions for Authors” this section should include the main conclusions that are missing.

Materials and Methods. Please consider next comments:

1)    Why a total of 980,383 SNPs were obtained from the OvineSNP600K BeadChip, which only contains 604,715 SNPs?

2)    Indicate in which version of the Sheep genome were the SNPs based on.

3)    Please explain more clearly which were the quality control standards used to select useful SNPs.

4)    Why the study did not include a test to adjust for multiple SNP comparisons such as Bonferroni adjustment test?

5)    Please explain the rationale to establish a significance threshold of –log10 (P) >5 for SNP significance.

Conclusions. This section appears to be a brief summary of the results. Instead, I suggest including 2 or 3 conclusive sentences describing the relevance of the study or the importance of the findings.

References. Please correct all references following guidelines described in “Instructions for Authors”.

Minor comments:

-       Line 29: Replace “xinjiang” by “Xinjiang”.

-       Line 37: Separate the square bracket form the text.

-       Line 43: Insert a conjunction word before “meat quality traits”.

-       Line 43: Replace “Single” by “single”.

-       Line 49: Separate the text after the period sign.

-       Lines 49-50: Please specify the traits that will be favored.

-       Line 56: Replace “are” by “were”.

-       Line 61: Replace “texel” by “Texel”.

-       Lines 68-70: These sentences were previously mentioned in lines 55-57.

-       Lines 78-79: This sentence is not clear, please rewrite.

-       Lines 85-86: These abbreviations should be stated in lines 79-82 before using.

-       Line 113: Insert the word “was” at the end of the sentence.

-       Line 115: Insert the word “where” at the beginning of the sentence.

-       Line 118: Insert a conjunction word before the last sentence.

-       Line 170: Separate the text from the period sign.

-       Line 173: Separate the text from the period sign.

-       Line 188: Separate the text from the period sign.

-       Line 233. Text after a period sign should start with a capital letter.

-       Lines 249-253: This paragraph is confusing, please correct or rewrite.

Reviewer 2 Report

Good paper with wide implications on meat quality. Here are some suggestions to improve the quality of this paper.

* Title of the paper needs to be more detailed. Instead of only pH, pH of meat or muscle might be the appropriate heading.

* Is sample size enough? What is the power of study?

* Can you provide correlation coefficients p values in line 86 or table 1 between phenotypes. 

* Can you provide PCA plot for this analysis?

* Please conduct pathway or network analysis on the results from GWAS. Also explain the results in biological context and possible effect on meat quality.

* What is the effect size or proportion of variance (pve) explained by these significant markers?

* Genes names need to be italicized.

* The tables and figures description needs to be detailed and self-explanatory without reading text. 

* Table 2 can be used as supplemental information.

* Conclusion needs to be more detailed with significance and implications of this study.

Reviewer 3 Report

1. The current version is hard to read.The etire manuscript needs to be reviewed for edits/English language/structure/grammar. Examples include but are not limited to:

(1) P1L10: to genotype of 462 Texel sheep×Altay sheep F2 individuals, and Genome-wide association analysis > to genotype 462 Texel sheep×Altay sheep F2 individuals, and genome-wide association analysis;

(2) P1L13: the longest dorsi musculature traits > the longissimus dorsi muscle traits;

(3) The results showed population stratification analysis found no significant systemic bias, and no obvious population stratification effect. this sentence makes me confused;

(4) P1L29: xinjiang > Xinjiang

(5) P1L43: Single > single;

(6) please unify the expression of pHtime thorough the manuscript;

(7) In this study, the pH values of longissimus dorsi muscle at 45min, 24h, 48h and 72h after slaughter were measured by using the sheep 600K chip technology to conduct GWAS study > In this study, the pH values of longissimus dorsi muscle at 45 min, 24 h, 48 h and 72 h after slaughter were measured and next using the sheep 600 K chip technology to conduct GWAS study;

(8) P2L56, P2L69, P3L123, P4L147, P5L152, P6L163, : are > were;

(9) Longissimus Dorsi muscle from the 12th to 13th ribs of the carcass of Texel sheep×Altay sheep F2 individuals were slaughtered. pH was measured 45min after slaughter, and the samples were stored in a refrigerator at 4℃ > Longissimus dorsi muscle from the 12th to 13th ribs of the carcass of Texel sheep×Altay sheep F2 individuals were collected. pH was measured after 45 min of slaughter, and the samples were stored in a refrigerator at 4 ℃;

(10) pH meter was measured by portable handheld pH meter purchased from TESTO, Germany > pH was measured by portable handheld pH meter purchased from TESTO, Germany;

(11) P3L92: 2ml > 2 mL;

(12) P3L103: include > included

(13) If the A260/280 ratio is 1.8-2.0 and A260/230 ratio is 1.8-2.1, and the concentration is greater than 100ng/μl, qualified DNA samples will be uniformly diluted to 50 ng/μl > Qualified DNA samples, A260/280 ratio in 1.8-2.0, A260/230 ratio in 1.8-2.1, concentration more than 100 ng/μl, will be uniformly diluted to 50 ng/μL;

(14) Two SNPS (OAR12_20003596) and SNPS (OAR12_20013208) located on chromosome  were … this sentence makes me confused. Additionally, SNPS or SNPs;

…………

…………

…………

2. please list references supporting your statements in P3L131;

3. In table 1: the minimum value was higher than the maximum; what is variable coefficient and how did the authors caculate it which was more than 1?

4. In Normal distribution analysis, was this step necessary for GWAS, and did the authors perform Normality Test with Kolmogorov-Smirnov or Jarque-Bera test?

5. Please provide the website of the Reference genome;

6. In Discussion,

(1) why there were so few loci associated with pH?

(2) what about the heritability of meat pH?

(3) Are there any similar studies reported previously?

(4) what about the result recurrence of this study compared with previous reports?

these information is suggested to be added in Discussion or Introduction.

Reviewer 4 Report

Comments: The manuscript entitled “Genome-wide association analysis of pH in Texel sheep × Altay sheep F2 resource population by Yi Long Zhao et al. has identify candidate genes associated with pH of mutton by Genome wide association analysis. The design of this study is of significance, however, only GWAS was performed in this study and the identified genes were not explored further. I do not think it is suitable for publication in the prestigious journal Animals.

In addition, I have some suggestions:

1. genes should be italicized;

2. the P-value should be italicized;

3. change " Longissimus Dorsi muscle" to " longissimus thoracis".